# Early Adolescents’ Motivations to Defend Victims of Cyberbullying

**DOI:** 10.3390/ijerph19148656

**Published:** 2022-07-16

**Authors:** Nathaniel Oliver Iotti, Damiano Menin, Tomas Jungert

**Affiliations:** 1Department of Psychology, Lund University, 222 41 Lund, Sweden; nathaniel_oliver.iotti@psy.lu.se; 2Department of Human Studies, University of Ferrara, 44121 Ferrara, Italy; damiano.menin@unife.it

**Keywords:** cyberbullying, bystander, defender, motivation to defend

## Abstract

The aim of this study was to investigate how different types of motivation to defend victims of bullying would be associated with various bystander behaviors in cyberbullying situations among early adolescents in Sweden. Data were collected from 460 Swedish adolescents aged between 11 and 15 years who completed a survey in their classroom. Results showed that autonomous motivation to defend was positively associated with defender behavior and negatively associated with pro-bully and passive behavior, while extrinsic motivation was positively associated with pro-bully and passive behavior. Age was positively associated with increased passive behavior and dampened defensive behavior, while no effect of gender was found on defender behavior. Our findings suggest that students’ autonomous motivation to defend victims is important in cyberbullying situations.

## 1. Introduction

Young adolescents today are increasingly connected with their peers and the rest of the world via their smartphones and social network memberships. Social media has become an important factor of adolescent life and affects the way youths spend their leisure time and maintain social relationships. In Sweden, 95% of all adolescents aged 13–16 have no less than one social media account, and 84% used social media every day in 2018 [1]. However, such extensive use of social media increases the likelihood of having unfavorable experiences, such as becoming victims or witnesses of cyberbullying.

### 1.1. Cyberbullying: Definition and Prevalence

Cyberbullying is defined as “An aggressive, intentional act carried out by a group or individual, using electronic forms of contact, repeatedly and over time against a victim who cannot easily defend him or herself.” [2], p. 376. However, more recently [3], the issues of repetition and power imbalance have been revised for the following reasons: repetition is not as important in cyberbullying as in traditional bullying because even a single act of aggression can be repeated several times and therefore have a snowball effect, and the power imbalance in this case relates more to different technical abilities when dealing with information technologies, as well as the possibility of anonymity [4].

The consequences of cyberbullying are well-documented [5] and victims experience a wide range of behavioral and emotional outcomes, including suicidal ideation, as well as attempts and in some cases completion [6,7]. Cyberbullying in Sweden is a serious problem, with one in every three students reporting that they have been exposed to some form of aggression online [8]. However, it should be mentioned that this datum is quite susceptible to change, depending on the definitions and measurement methods employed [9].

### 1.2. Participant Roles

Cyberbullying does not happen in a vacuum, but it is a situation that involves bystanders. According to research, bystanders can take on a variety of different roles when witnessing bullying or cyberbullying. Said roles range from defending the victim, to being passive and not intervening, to supporting the bully [10]. Research has shown that bullying is less frequent in schools where defending behaviors are more prevalent and pro-bullying behaviors are less prevalent [11,12], and recent research has shown that defending behavior also plays an important role in online settings [13]. Therefore, defending behaviors can be considered very important in reducing bullying. However, defending behaviors remain quite rare, compared to other behaviors. The reasons for this are multiple but can conceivably be reduced to the fact that defending a victim is a potentially dangerous task with an unknown outcome. Potential defenders might fear that they will be bullied themselves, decrease in social status, and perhaps get hurt, just to name a few possibilities. Research has investigated a series of aspects that play a role in influencing who becomes a defender, and they are: (a) feelings of self-efficacy [14], (b) motivation to defend [15], (c) student-teacher relationships [16], and (d) moral emotions and moral disengagement [17]. For the purpose of this study, we have chosen to focus on motivation to defend victims, as it plays a fundamental role in influencing defending behaviors [15].

### 1.3. Motivation and Defending Victims

According to Self-Determination Theory (SDT) [18], motivation spans along a continuum, from amotivation, or the lack of motivation, to identified motivation, which is the most internal form of motivation. There are four types of motivation, listed here from most extrinsic to most intrinsic: external, introjected, integrated and identified. Often, external and introjected motivation are collapsed into the term ‘controlled motivation’, and integrated and identified motivation are collapsed into the term ‘autonomous motivation’. Controlled motivation entails acting with a sense of pressure, or of being required to involve oneself in activities that one might not otherwise pursue, while autonomous motivation entails acting with a sense of choice and of having some options available. These types of motivation differ in their fundamental regulatory processes and accompanying experiences: while autonomous motivation is a natural, inherent drive to seek out challenges and new possibilities that is associated with cognitive and social development, controlled motivation comes from external sources. Individuals who are autonomously motivated freely engage in an interesting activity simply for the enjoyment and excitement it brings, rather than to get a reward or to satisfy a certain constraint. Conversely, controlled motivation is instrumental in nature: behaviors are not performed out of interest, but for the consequences that they are thought to be instrumentally linked to. It should be noted that controlled motivation is not seen as negative per se, but it is seen as regulated by external contingencies and therefore more prone to poor maintenance and transference once the contingencies have been removed. Consequently, it is not a type of motivation that is considered optimal because of its high volatility. Conversely, autonomous motivation is more internalized and can be considered the most optimal sort of motivation because the underlying value of an action is accepted and shared by the agent. Defending other people is a type of prosocial behavior, which has been associated with autonomous motivation [19]. Jungert et al. [15], in a study on traditional bullying, found that autonomous motivation to defend victims was positively associated with defending and negatively associated with passive bystanding, whereas extrinsic motivation to defend was positively associated with pro-bullying behaviors (i.e., assisting the bullies or reinforcing the bullying). Nevertheless, there is still a lack of research on how young adolescents’ motivation to defend victims of cyberbullying works and what it is influenced by.

### 1.4. Gender and Age Differences

Finally, some studies found girls to be more inclined than boys to defend victims of cyberbullying [20,21] and that helpful bystander behaviors are likely to decline with age [22], with younger students being more motivated to defend victims than older students. However, said findings are mixed [22,23] and require more investigation.

### 1.5. The Present Study

Based on previous findings, our goal was to examine differences in early adolescents’ cyberbullying roles and self-determined motivation to defend victims during cyberbullying incidents between boys and girls and depending on age. Our research questions were the following:Will autonomous motivation to defend positively predict defending behaviors, and negatively predict pro-bullying behaviors and passive bystanding?Will extrinsic motivation to defend positively predict pro-bullying behaviors and passive bystanding, and negatively predict defending behaviors?

In addition to the above research questions, associations between age and gender and motivation were examined, as well as interaction effects in a more exploratory fashion, as these interactions had not been studied previously.

More specifically, concerning age, we wanted to investigate whether we could confirm previous findings [22] that defending behaviors appear to decrease as students get older. Concerning gender differences, we also wanted to explore whether our data supported findings of girls being more autonomously motivated to defend victims than boys [20,21].

Our study provides a significant contribution to the literature on this topic because, by combining a Self-Determination-Theory-based understanding of motivation with current knowledge of cyberbullying roles and processes, it helps fellow scholars and practitioners obtain a better grasp of a phenomenon where much remains to be disentangled, which is young adolescents’ motivation to defend victims of bullying. Furthermore, our study aims to enrich the conversation on defensive behaviors in cyberbullying by providing meaningful findings and stimulating insight on an issue that remains relevant and that has very concrete implications for both young adolescents and the adults that care for them.

## 2. Materials and Methods

### 2.1. Participants and Procedure

Participants were enrolled in 24 classes in grades 4–8 in six elementary schools in Sweden. The Swedish school system begins with preschool at the age of six and is followed by primary school (grades 1–6) and secondary school (grades 7–9) of compulsory schooling. The six schools that were included in the study were all schools in a large Swedish city. Convenience sampling was conducted in each school with the assistance of class teachers. Parental consent letters were distributed to the families of all children in these classes and the study was authorized by the school administration. The respondents did not receive any reimbursement for their participation. Overall, 460 students completed the survey (57% female) aged between 11 and 15 years (M = 11.80; SD = 1.08). The students had the following grade distribution: 15% fourth grade (70 students), 42% fifth grade (193 students), 27% sixth grade (127 students), 6% seventh grade (26 students), and 10% eighth grade (44 students).

Data collection was carried out with pen-and-paper surveys during class time. Students were given the opportunity to ask questions and were informed that their participation would be voluntary and anonymous, should they decide to join the study. Furthermore, students were informed that they had the right to withdraw from the study at any time. The students who had not received consent from their parents, or did not wish to participate themselves, were assigned a different task from their teachers. Participants were asked to answer questions truthfully, and survey completion took between 10 and 20 min. Finally, data collection was carried out during April/May 2020.

### 2.2. Materials

#### 2.2.1. Motivation to Defend Scale (MDS)

The Motivation to Defend Scale [24], which comprises 14 items, assessed the students’ motivation to defend victims during situations of school bullying. The items are divided into four subscales (i.e., Extrinsic, Introjected, Identified, and Intrinsic), each measuring a different motivational type. The Extrinsic motivation subscale consists of five items, while all other subscales consist of three items each. Before filling out the scale, participants are asked to think back on episodes in which they had witnessed other students being bullied, and to report why they would want to help victims.

Example items include “Because I like to help other people” (intrinsic motivation), “Because I think it is important to help people who are treated badly” (identified regulation), “Because I would feel like a bad person if I did not try to help” (introjected regulation), and “To become popular” (extrinsic motivation). Participants answer each item on a 5-point scale ranging from 1 (“*completely disagree*”) to 5 (“*completely agree*”). In line with previous literature [25], the intrinsic and identified subscales were merged to form an autonomous subscale.

#### 2.2.2. Participant Role Scale (PRS)

Participant roles were measured with an adapted Swedish version of a 15-item self-report scale [14], which examined participants’ tendency to fit various cyberbullying-related profiles during a school year. The roles were pro-bully (who either actively assist bullies or cheer them on), passive bystander (not participating in bullying behavior, but not stopping it), and defender (actively supporting a victim). The items evaluating these profiles included “If I see that another student has been teased with nasty messages on the mobile or internet, I give a “thumbs up” or otherwise “like” the messages.” (pro-bullying), “I did nothing special, but was passive.” (passive bystanding), and “I tried to get the bully/bullies to stop by telling them in some way.” (defending). Participants answer each item on a 5-point scale ranging from 1 (“*completely disagree*”) to 5 (“*completely agree*”).

### 2.3. Ethics

The study was ethically approved by the Swedish Ethics Review Authority on 24 January 2020. Both parental and individual informed consent were obtained from students and their families before allowing them to participate in the study. Furthermore, participants did not receive any form of retribution or compensation for their involvement in the study. Finally, the study complied with the principles expressed in the Declaration of Helsinki, and participation was anonymous and voluntary.

### 2.4. Data Analysis

Confirmatory Factor Analysis (CFA) was used to assess the goodness of the factor structure, and Structural Equation Modeling (SEM) was employed to test a model including gender and age as exogenous variables, autonomous and extrinsic motivation as mediators and passive behavior, defender role and pro-bully attitude as outcomes, using the distributionally robust maximum likelihood estimator. All analyses were carried out in the R statistical environment, version 4.0.2 [26] using the lavaan package [27].

## 3. Results

A CFA was performed in order to test the measurement model, highlighting an adequate fit: CFI = 0.920, TLI = 0.900, RMSEA = 0.051, SRMR = 0.051. Items with loadings lower than 0.5 were removed (item 3 from the extrinsic motivation scale and item 2 from the autonomous motivation scale). Factor loadings were all significantly different from zero.

As shown in Table 1, the CFA highlighted significant covariances in the expected directions (*p* < 0.001) for all the study variables.

Next, the hypothesized structural model was fitted, showing an acceptable fit: CFI = 0.918, TLI = 0.895, RMSEA = 0.048, SRMR = 0.049. SEM analysis highlighted a strong positive association between autonomous motivation and defender behavior (β = 0.782, *p* < 0.001) and negative associations between autonomous motivation and pro-bully attitude (β = −0.377, *p* < 0.001) and passive behavior (β = −0.493, *p* < 0.001). Extrinsic motivation, conversely, was positively associated with pro-bully attitude (β = 0.249, *p* = 0.003) and passive behavior (β = 0.235, *p* = 0.003). No effect of extrinsic motivation on defender behavior was found (β = −0.090, *p* = 0.219). Older age was found to be associated with increased passive behavior (β = 0.196, *p* = 0.001) and dampened defensive behavior (β = −0.166, *p* = 0.008). Girls were less likely to show extrinsic motivation (β = −0.139, *p* = 0.015) but the total effects of gender on pro-bully attitude (β = 0.050, *p* = 0.300) and passive behavior (β = 0.046, *p* = 0.365) were not significant, while no effect of gender was found on defender behavior. The model is depicted in Figure 1.

## 4. Discussions

There seems to be a knowledge gap regarding early adolescents’ cyberbullying roles and self-determined motivation to defend victims during cyberbullying. To our knowledge, the present study was the first to examine whether different types of motivation to defend victims of cyberbullying would be associated with different bystander roles in cyberbullying and if this would be associated with age and gender differences.

### 4.1. Motivation and Bystander Behaviors

The two research questions were confirmed and motivation to defend, similarly to traditional bullying, works as anticipated, based on previous literature. The present findings, in fact, showed that autonomous motivation was positively associated with greater defending behavior while extrinsic motivation was positively associated with pro-bully attitudes and passive behavior. In addition, autonomous motivation was negatively associated with passive bystanding and pro-bully attitudes. Our findings can be compared with previous research showing that autonomous motivation is related to greater defending [15,25]. This is in line with self-determination theory [28] and with prior research demonstrating that autonomous motivation is associated with stronger persistence and performance in other activities [29,30,31,32], including prosocial behavior [19]. In general, our results suggest the importance of having a high autonomous motivation to defend victims of cyberbullying, as the relationship between autonomous motivation and defender behavior was particularly strong.

Furthermore, the findings revealed that extrinsic motivation was linked to greater passive bystanding, while autonomous motivation was negatively associated with this bystander response. The motivation type that had the strongest association with passive bystanding in the model was extrinsic motivation, which suggests that individuals who only or mostly consider defending a victim of cyberbullying if it would benefit themselves (i.e., attain social rewards or avoid social punishments) would be those who are most inclined to remain as passive bystanders when witnessing cyberbullying. Considering that bullies are usually powerful and high-status peers [33], fear of social punishments in terms of being victimized before others because of interfering [34,35] might make these students more prone to remain passive. In contrast, students high in autonomous motivation might be less concerned with social costs as possible outcomes of defending the victim.

We also found that greater extrinsic motivation and less autonomous motivation were associated with greater pro-bullying attitude. The positive association between extrinsic motivation and pro-bullying and passive behavior might be due to the fact that extrinsically motivated students are more influenced by how they interpreted and anticipated social rewards and social sanctions, which might vary across bystander situations and contexts. Students who are high in extrinsic motivation might be those who respond most in accordance with the *arousal: cost–reward bystander model* [36], as they appear to be those who are most concerned predicting costs and rewards for different bystander behaviors and decide to behave in ways that they believe maximize social rewards and minimize social sanctions. Extrinsic motivation was also linked to passive behavior, which supports our hypothesis that students who are highly externally regulated in bystander situations may be predisposed to see passive bystanding as the safest course of action, because it maximizes punishment avoidance in risky situations. Very extrinsically motivated students might believe that staying passive is the most efficient way of avoiding being bullied themselves. In addition, they might reason that not getting involved could help them avoid social backlash from most, or popular, non-bullying peers and to evade disciplinary consequences, or reports to their parents, which might be considered a risk of openly engaging in pro-bullying behaviors.

### 4.2. Age, Gender, and Bystander Behaviors

The analysis regarding bystander behavior and age showed that older students were more inclined to remain passive and less likely to defend. Thus, age seems to play a role in bystander behaviors where younger bystanders take a clearer stand in defending the victim of cyberbullying. The finding that autonomous motivation declines with age is in line with previous studies that have found a gradual deterioration of intrinsic academic motivation across the school years [37,38]. According to these findings, younger students appear to be more represented in groups that are characterized by higher intrinsic and identified motivation, and lower extrinsic motivation, while the opposite appears to be true for older students. This difference seems to hold when we consider motivation to defend victims of bullying. It has been reported that helpful bystander behaviors tend to decrease with age [39,40], while extrinsic motivation appears to be higher at 14 years of age [23]. This could be linked to findings that suggest that bullying behaviors peak around age 14 and that adolescents in that age group might have stronger pro-bullying attitudes [12].

Moreover, the analysis showed that girls were less likely to show extrinsic motivation to defend. However, girls were not more prone to pro-bullying behaviors or passive bystanding. Thus, we did not receive any support for gender differences in bystander behavior. This is in line with a previous study that found that female and male bystanders were relatively evenly distributed across different profiles of motivation to defend [23]. According to previous research, there has been a tendency to overrepresent females in autonomous motivation groups, while underrepresenting them in controlled motivation groups. However, these supposed gender differences do not seem to hold when we look at prosocial motivation to defend victims of cyber bullying.

### 4.3. Limitations

Some limitations of this study should be mentioned. The data were collected with self-report measures, which are sensitive to social desirability, perception and recall biases, and shared method variance effects. Furthermore, a cross-sectional study design was adopted; therefore, we were unable to determine the direction of effects between the variables. It is possible that the associations found in the study are reciprocal. Future research needs to take a longitudinal approach to examine directionality, including possible bidirectional relationships, among the study variables.

Another possible limitation is that the bystander scale assumes that the students have seen or observed cyberbullying, which might not always be the case. On the other hand, it would be unlikely that students never had witnessed such negative behavior.

Finally, a note of caution needs to be sounded regarding the generalization of the findings, as the study sample consisted of 460 students from rural areas to large cities in Sweden. Future studies might expand on the current findings by considering students in other countries.

### 4.4. Implications for Practice

These limitations aside, the current findings have implications. Practitioners are recommended to carefully assess motivations underlying not only bullying but also bystander behaviors of friends and peers of students involved in cyberbullying. Over the years, scholars have recognized the salience of interventions involving children who are bystanders or witnesses in bullying situations [41]. Findings from our study also confirm that bystander interventions are of critical importance to students who are involved in cyberbullying in Sweden. Considering that cyberbullying is a group phenomenon, parents and practitioners in school settings alike are advised to help increase bystanders’ actions in cyberbullying situations, and one way to do so is to increase bystanders’ involvement in the existing bullying prevention programs [41].

### 4.5. Future Directions

In the future, it would be important to investigate the bases for autonomous and extrinsic motivation to defend, in terms of traits or processes that lead to a tendency to have an extrinsic or tendentially autonomous motivation. In such research there could, and should, be a focus on how parenting styles and broader family dynamics can influence bystander behaviors, as this is still a topic that requires further investigation, and that can provide us with a different angle to view cyberbullying processes from. Having a more comprehensive picture of what contributes to the behaviors of all the actors involved in a cyberbullying episode would provide clear benefits; for example, it would help practitioners to design more effective interventions aimed at tackling this very real and present societal problem.

## 5. Conclusions

This study found similarity in motivational processes between traditional bullying and cyberbullying. Our results showed that autonomous motivation to defend was positively associated with defender behavior and negatively associated with pro-bully and passive behaviors, while extrinsic motivation was positively associated with pro-bully and passive behaviors. Furthermore, age was positively associated with increased passive behavior and dampened defensive behavior, while no effect of gender was found on defender behavior.

Overall, our results appear to confirm that online interactions can also be contextualized to the psychosocial dynamics that concern traditional bullying and, therefore, do not exist in a vacuum and must be considered in their social and group facets, especially concerning defender motivation. Finally, this study supports the importance of involving bystanders in prevention programs, as they play an essential role in helping decrease and tackle bullying and cyberbullying episodes.

## Figures and Tables

**Figure 1 ijerph-19-08656-f001:**
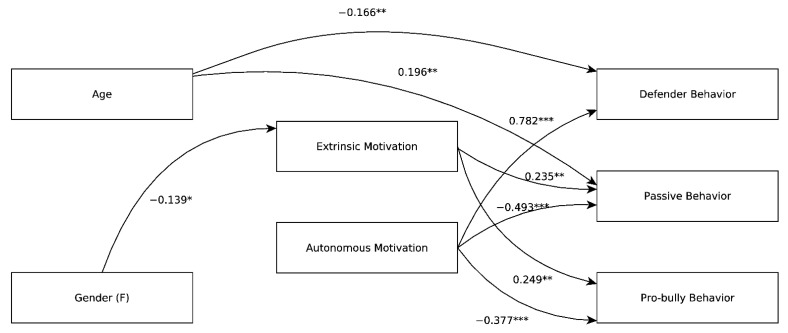
The structural model. ***: *p* < 0.001; **: 0.001 < *p* < 0.01; *: 0.01 < *p* < 0.05; only significant effects (*p* < 0.05) are shown.

**Table 1 ijerph-19-08656-t001:** CFA covariance matrix.

	EM	AM	PA	PB	DR
Extrinsic Motivation (EM)	--	−0.219 ***	0.321 ***	0.320 ***	−0.246 ***
Autonomous Motivation (AM)		--	−0.425 ***	−0.557 ***	0.811 ***
Pro-bully Attitude (PA)			--	0.640 ***	−0.499 ***
Passive Behavior (PB)				--	−0.646 ***
Defender Role (DR)					--

*** *p* < 0.001.

## Data Availability

Data that support the findings of the study are available from the corresponding author upon reasonable request.

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
