# Peer review of "Early Adolescents’ Motivations to Defend Victims of Cyberbullying"

_ijerph, 2022, doi:10.3390/ijerph19148656_

Round 1
Reviewer 1 Report
This study examined the relationship between different types of motivation and various bystander behaviors in cyberbullying situations. It is significant in both theory and practice. In sum, the method, data analysis, results and discussion of this study are reasonable. But there exist several issues that should be further considered.
1. Age and gender are also variables concerned by the authors, and assumptions can also be given in the research questions.
2. The reliability of the measurements is not reported.
3.I wonder how the participant role is classified, and the number of participants of each type should be reported.
Author Response
Response to Reviewer 1 Comments
This study examined the relationship between different types of motivation and various bystander behaviors in cyberbullying situations. It is significant in both theory and practice. In sum, the method, data analysis, results and discussion of this study are reasonable. But there exist several issues that should be further considered.
Point 1: Age and gender are also variables concerned by the authors, and assumptions can also be given in the research questions.
Response 1: We thank the reviewer for kindly pointing this out, and we have now added more explicit assumptions on age and gender in the appropriate section (please see lines 111-115).
Point 2: The reliability of the measurements is not reported.
Response 2: We fully share the reviewer's concern about the psychometric properties of measured constructs. For this very reason, we adopted Confirmatory Factor Analysis to verify the factor structure of our measurement system. This method is widely regarded as a more advanced and appropriate approach compared to, for example, Cronbach's alpha (see, e.g., Flora, 2020: https://doi.org/10.1177%2F2515245920951747). However, if the reviewer or the editor think that we should add additional information about reliability (e.g., Cronbach's alpha or Omega coefficient) we will be happy to comply.
Point 3: I wonder how the participant role is classified, and the number of participants of each type should be reported.
Response 3: We thank the reviewer for this question. We would like to clarify that, in this study, the Participant Role Scale was used to estimate individual scores for each subscale, and not to classify participants, as this approach would have resulted in losing a significant amount of information. The instrument itself was never designed to classify respondents based on their roles, but to measure their inclination towards different bystander behaviors, and, therefore, it lacks pre-specified cutoffs for its subscales. This is the reason why the prevalence rates of the different roles were not reported.
Reviewer 2 Report
I enjoyed the opportunity to review this manuscript on an important topic that warrants additional research and scholarship. This manuscript investigated how different types of motivation to defend victims in bullying would be associated with various bystander behaviors in cyberbullying situations among early adolescents in Sweden. This paper is well-written and the authors have taken care in its presentation. The purpose is clearly stated and the literature review is comprehensive with the use of relevant and up to date literature. The methods section is transparent and explicit, providing sufficient details to allow the work to be reproduced. Interpretations are appropriate and justified in relation to analyses. Additionally, the authors acknowledged several limitations associated with the study. With the aim of providing feedback to help improve the paper further, the following minor comments regarding the method section are provided below:
1. Include when the data was collected (e.g., What month/year)
2. Include the total enrollment number for grades 5-8 of the 6 elementary schools?
3. “Students who had not been given consent from their parents, or did not want to participate, received a different task from their teachers.” What was the percentage of this??
Thank you for the opportunity to review this article.
Author Response
Response to Reviewer 2 Comments
I enjoyed the opportunity to review this manuscript on an important topic that warrants additional research and scholarship. This manuscript investigated how different types of motivation to defend victims in bullying would be associated with various bystander behaviors in cyberbullying situations among early adolescents in Sweden. This paper is well-written and the authors have taken care in its presentation. The purpose is clearly stated and the literature review is comprehensive with the use of relevant and up to date literature. The methods section is transparent and explicit, providing sufficient details to allow the work to be reproduced. Interpretations are appropriate and justified in relation to analyses. Additionally, the authors acknowledged several limitations associated with the study. With the aim of providing feedback to help improve the paper further, the following minor comments regarding the method section are provided below:
Point 1: Include when the data was collected (e.g., What month/year)
Response 1: We have now added this to the manuscript (please see line 150).
Point 2: Include the total enrollment number for grades 5-8 of the 6 elementary schools?
Response 2: This has now been added to the manuscript (please see lines 140-142).
Point 3: “Students who had not been given consent from their parents, or did not want to participate, received a different task from their teachers.” What was the percentage of this??
Response 3: No participant requested a different task from their teacher because they were either absent from school on the day of data collection or they had given their consent to take the survey. This is why we did not include a percentage, as it would have been 0%.
Reviewer 3 Report
Thank you for inviting me to review this paper, which is a very interesting and well-organized study. However, I have some suggestions that might help strengthen the manuscript that I will make to the authors.
Introduction:
· The section could be enriched with the role of the family in bullying and cyberbullying behaviors. There is abundant scientific literature on this subject, some examples of articles that may help to expand on this section are as follows:
Martínez, I., Murgui, S., Garcia, O.F., & Garcia, F. (2019b). Parenting in the digital era: Protective and risk parenting styles for traditional bullying and cyberbullying victimization. Computers in Human Behavior, 90, 84-92.
Navarro, R., Serna, C., Martínez, V., & Ruiz-Oliva, R. (2013). The role of Internet use and parental mediation on cyberbullying victimization among Spanish children from rural public schools. European Journal of Psychology of Education, 28(3), 725-745.
The present study
· Why do the authors only formulate research questions and not hypotheses? However, in the discussion the authors say that both hypotheses were confirmed. It is advisable to formulate hypotheses clearly.
· This paragraph needs to be further clarified: “To our knowledge, our contribution seems to be the first of its kind, and it is relevant because it confirms and expands upon previous findings and contributes to shedding light on the influence of motivation on bystander roles in cyberbullying”. We should be careful with this type of statements if they are not well corroborated or better specify in which aspects the study is novel with respect to others. In fact, there is indeed scientific literature on the role of bystander motivation in the phenomena of bullying and cyberbullying. It is advisable to soften or better define this statement.
Materials and Methods
Participants and Procedure
· Overall, 460 students completed 123 the survey (57% female) aged between 11 and 15 years (M = 11.80; SD = 1.08): The mean age seems to be unbalanced downwards. Please check the age distribution in case there is an error.
· Participants were enrolled in 24 classes in grades 5–8 in six elementary schools in Sweden… The students had the following grade distribution: 15% fourth grade, 42% fifth grade, 27% sixth grade, 6% seventh grade, and 10% eighth grade: First, the authors say that the participants were students in grades 5 to 8, but then point out that 15% were in grade 4. If this is a typo they should correct it, perhaps that is why the mean age seems low. Anyway, could they explain why the distribution by grade is so unbalanced?
· It is recommended that the information be better organized, first describing the sample and then the procedure, since everything is mixed up.
Conclusions
· This section should be expanded and extended a little more.
· Finally, the sentence "the section is mandatory" may be a typo.
Author Response
Response to Reviewer 2 Comments
Introduction:
Point 1: The section could be enriched with the role of the family in bullying and cyberbullying behaviors. There is abundant scientific literature on this subject, some examples of articles that may help to expand on this section are as follows:
Martínez, I., Murgui, S., Garcia, O.F., & Garcia, F. (2019b). Parenting in the digital era: Protective and risk parenting styles for traditional bullying and cyberbullying victimization. Computers in Human Behavior, 90, 84-92.
Navarro, R., Serna, C., Martínez, V., & Ruiz-Oliva, R. (2013). The role of Internet use and parental mediation on cyberbullying victimization among Spanish children from rural public schools. European Journal of Psychology of Education, 28(3), 725-745.
Response 1: We thank the reviewer for this suggestion. After some careful consideration, we have decided against expanding the introduction by discussing the role of the family in bullying and cyberbullying behaviors, as we fear that it would somewhat distract readers from the focus of our study, which is on bystanders, and their roles and motivation to defend victims of cyberbullying. Given that studies on the role of families in bullying and cyberbullying mostly focus on victims or perpetrators, it would not enrich our narrative to add a section on this. However, we agree that research on families and their possible influence on bystander behaviors is very much needed and have discussed this in the Future directions section (please see lines 326-333).
The present study
Point 2: Why do the authors only formulate research questions and not hypotheses? However, in the discussion the authors say that both hypotheses were confirmed. It is advisable to formulate hypotheses clearly.
Response 2: We thank the reviewer for pointing this out to us. We prefer to use research questions to frame our study, and we have now edited the manuscript to reflect this, removing any reference to hypotheses and discussing research questions instead.
Point 3: This paragraph needs to be further clarified: “To our knowledge, our contribution seems to be the first of its kind, and it is relevant because it confirms and expands upon previous findings and contributes to shedding light on the influence of motivation on bystander roles in cyberbullying”. We should be careful with this type of statements if they are not well corroborated or better specify in which aspects the study is novel with respect to others. In fact, there is indeed scientific literature on the role of bystander motivation in the phenomena of bullying and cyberbullying. It is advisable to soften or better define this statement.
Response 3: We thank the reviewer for inviting us to redefine our statement. We have now edited the paragraph (please see lines 116-124) to better specify the novelty of our contribution and its relevance.
Materials and Methods
Participants and Procedure
Point 4: Overall, 460 students completed 123 the survey (57% female) aged between 11 and 15 years (M = 11.80; SD = 1.08): The mean age seems to be unbalanced downwards. Please check the age distribution in case there is an error.
Response 4: We thank the reviewer for pointing this out. We can confirm that there is no error in the age distribution of our sample and that it is, indeed, unbalanced downward. This is due to convenience sampling. However, we decided to keep all ages because, as per our research questions, we were interested in investigating possible decreases in defending behaviors among older students. Our reasoning was that even a small finding on this might be an interesting contribution to the literature, so we decided to keep older participants in our analyses.
Point 5: Participants were enrolled in 24 classes in grades 5–8 in six elementary schools in Sweden… The students had the following grade distribution: 15% fourth grade, 42% fifth grade, 27% sixth grade, 6% seventh grade, and 10% eighth grade: First, the authors say that the participants were students in grades 5 to 8, but then point out that 15% were in grade 4. If this is a typo they should correct it, perhaps that is why the mean age seems low. Anyway, could they explain why the distribution by grade is so unbalanced?
Response 5: We thank the reviewer for spotting this typo. Participants were students in grades 4-8 and this has now been corrected in the manuscript. Concerning the grade distribution imbalance, please see the above reply.
Point 6: It is recommended that the information be better organized, first describing the sample and then the procedure, since everything is mixed up.
Response 6: We thank the reviewer for this observation. We have reorganized the appropriate section by adding a clearer distinction between sample description and study procedure.
Conclusions
Point 7: This section should be expanded and extended a little more.
Response 7: We thank the reviewer for this suggestion. We have now expanded the section (please see lines 336-347).
Point 8: Finally, the sentence "the section is mandatory" may be a typo.
Response 8: We thank the reviewer for spotting this typo. We have now removed the sentence.